# Social Inequalities in Mental Health and Self-Perceived Health in the First Wave of COVID-19 Lockdown in Latin America and Spain: Results of an Online Observational Study

**DOI:** 10.3390/ijerph20095722

**Published:** 2023-05-04

**Authors:** Carmen Salas Quijada, Natalia López-Contreras, Tomás López-Jiménez, Laura Medina-Perucha, Brenda Biaani León-Gómez, Andrés Peralta, Karen M. Arteaga-Contreras, Anna Berenguera, Alessandra Queiroga Gonçalves, Olivia Janett Horna-Campos, Marinella Mazzei, Maria Sol Anigstein, Jakeline Ribeiro Barbosa, Olga Bardales-Mendoza, Joan Benach, Daiane Borges Machado, Ana Lucía Torres Castillo, Constanza Jacques-Aviñó

**Affiliations:** 1Facultad de Medicina, Universidad Austral de Chile, Valdivia 5110566, Chile; carmen.salas@uach.cl; 2Vicerrectoría de Investigación y Postgrado, Universidad de La Frontera, Temuco 4811230, Chile; natalia.lopez@ufrontera.cl; 3Fundació Institut Universitari per a la Recerca a l’Atenció Primària de Salut Jordi Gol i Gurina (IDIAPJGol), 08007 Barcelona, Spain; tlopez@idiapjgol.org (T.L.-J.); lmedina@idiapjgol.org (L.M.-P.); aberenguera@idiapjgol.org (A.B.); 4Universitat Autònoma de Barcelona, 08193 Cerdanyola del Vallès, Spain; 5Network for Research on Chronicity, Primary Care, and Health Promotion (RICAPPS), Spain; aqueiroga@idiapjgol.info; 6Unitat de Suport a la Recerca Metropolitana Nord, Fundació Institut Universitari per a la Recerca a l’Atenció Primària de Salut Jordi Gol i Gurina (IDIAPJGol), 08303 Mataró, Spain; bblego@gmail.com; 7Public Health Institute, Pontificia Universidad Católica del Ecuador (PUCE), Quito 170525, Ecuador; tirico85@gmail.com (A.P.); atorres331@puce.edu.ec (A.L.T.C.); 8Servicios de Atención Psiquiátrica, Anillo Periférico #2767, Ed.5 P.B., Alcaldía La Magdalena Contreras, Cuidad de México 10200, Mexico; arteaga.med@gmail.com; 9Departament d’Infermeria, Universitat de Girona, Emili Grahit, 77, 17003 Girona, Spain; 10Unitat de Suport a la Recerca Terres de l’Ebre, Fundació Institut Universitari per a la Recerca a l’Atenció Primària de Salut Jordi Gol i Gurina (IDIAPJGol), 43500 Tortosa, Spain; 11Unitat Docent de Medicina de Família i Comunitària Tortosa-Terres de L’Ebre, Institut Català de la Salut, 43500 Tortosa, Spain; 12Escuela de Salud Pública “Salvador Allende”, Facultad de Medicina, Universidad de Chile, Santiago 8380000, Chile; oliviahorna@uchile.cl (O.J.H.-C.); mmazzei@med.uchile.cl (M.M.); msanigste@uchile.cl (M.S.A.); 13Departamento de Antropología, Universidad de Chile, Santiago 6850331, Chile; 14Center for Epidemiology and Health Surveillance, Oswaldo Cruz Foundation, Brasília 70904-130, Brazil; jakelinebarbosa@gmail.com; 15Facultad de Educación, Universidad Peruana Cayetano Heredia, Lima 15102, Peru; olga.bardales.m@upch.pe; 16Research Group on Health Inequalities, Environment, and Employment Conditions (GREDS-EMCONET), Universitat Pompeu Fabra, 08002 Barcelona, Spain; joan.benach@upf.edu; 17Johns Hopkins University-Universitat Pompeu Fabra Public Policy Center (UPF-BSM), 08002 Barcelona, Spain; 18Ecological Humanities Research Group (GHECO), Universidad Autónoma de Madrid, 28049 Madrid, Spain; 19Center of Data and Knowledge Integration for Health, Gonçalo Moniz Institute, Oswaldo Cruz Foundation, Salvador 41745-715, Brazil; daianedbm@hotmail.com; 20Department of Global Health and Social Medicine, Harvard Medical School, Boston, MA 02115, USA

**Keywords:** COVID-19, social impact, lockdown, mental health, inequities, self-perceived health

## Abstract

COVID-19 lockdowns greatly affected the mental health of populations and collectives. This study compares the mental health and self-perceived health in five countries of Latin America and Spain, during the first wave of COVID 19 lockdown, according to social axes of inequality. This was a cross-sectional study using an online, self-managed survey in Brazil, Chile, Ecuador, Mexico, Peru, and Spain. Self-perceived health (SPH), anxiety (measured through GAD-7) and depression (measured through PHQ-9) were measured along with lockdown, COVID-19, and social variables. The prevalence of poor SPH, anxiety, and depression was calculated. The analyses were stratified by gender (men = M; women = W) and country. The data from 39,006 people were analyzed (W = 71.9%). There was a higher prevalence of poor SPH and bad mental health in women in all countries studied. Peru had the worst SPH results, while Chile and Ecuador had the worst mental health indicators. Spain had the lowest prevalence of poor SPH and mental health. The prevalence of anxiety and depression decreased as age increased. Unemployment, poor working conditions, inadequate housing, and the highest unpaid workload were associated with worse mental health and poor SPH, especially in women. In future policies, worldwide public measures should consider the great social inequalities in health present between and within countries in order to tackle health emergencies while reducing the health breach between populations.

## 1. Introduction

In order to contain the transmission of COVID-19, countries implemented measures such as movement restrictions with a strong impact on transportation, food security, the economy, and access to healthcare and education [1]. These drastic changes had psychological repercussions, bringing with them emotional conflicts, depression, stress, insomnia, and changes in health behaviors [2,3]. These symptoms varied according to the social and material conditions of people’s lives and according to social axes of inequality such as gender, age, territory, or socioeconomic position [4,5,6]. Likewise, women have been the most relevant workforce in health services that have contained the pandemic, a labor sector with the worst results in mental health [7]. In Latin America (LA), women are more present in the informal sector, have less capacity to deal with socioeconomic problems, and are the main caregivers at home [8]. In LA countries, a worse quality of life [9] and greater emotional distress have been observed among young people [10], especially women [11], as an effect of COVID-19 lockdown. In addition, self-perceived health (SPH)—an indicator of people’s health, the use of health services, and mortality [12,13]—has worsened, especially among older women [14].

LA is one of the most unequal regions in the world, with large cultural, social, and economic differences between and within countries [15]. In 2020, the Gini index of inequality was 48.9 in Brazil, 45.4 in Mexico, 47.3 in Ecuador, 44.9 in Chile, and 43.8 in Peru, showing high inequality compared to 34.3 in Spain [16]. Currently, LA has been one of the regions most affected by the COVID-19 pandemic in the world, with significant contractions in the middle-income population and an increase in inequality and poverty that has affected women, young people, migrants, and less educated workers the most [17]. The high level of informal work and the inability of governments to provide socioeconomic support for basic subsistence made compliance with the restrictions even more difficult [18]. For its part, in Spain, Temporary Employment Regulation Records (ERTEs) were applied (70% of salary), especially in activities classified as “nonessential”, which affected women’s employment more and deepened the wage gap [19]. In LA, most countries closed educational establishments for at least three semesters, while Spain closed for 4 months [20]. Moreover, Peru was among the strictest in home containment, and Brazil was among the most lax [15]. On the other hand, the pandemic arrived in the midst of strong social conflicts, with Chile, Peru, and Ecuador were immersed in a major sociopolitical crisis, questioning government institutions [15].

This diversity of contexts suggests the existence of a differentiated impact on mental health and SPH between LA countries and Spain. The latter has a more developed welfare state, but relatively similar sociocultural characteristics to LA. For this reason, Spain and five LA countries were studied: Brazil, Chile, Ecuador, Mexico, and Peru. This information is particularly relevant for the development of strategies for targeting resources to the most vulnerable populations, as well as creating policies and developing public strategies.

### 1.1. Brief Socio-Sanitary Context

A brief contextualization of the countries analyzed is given below.

#### 1.1.1. Brazil

The health emergency caused by the COVID-19 pandemic was marked by an adverse political context. In general, the federal executive (Bolsonaro was president at the time) was against the adoption of the main measures recommended by experts, such as the use of masks and social isolation. Brazil never adopted a total shutdown of the country; social isolation was carried out irregularly. In this scenario, Brazilian society experienced an intensification of the political polarization already present since the last presidential elections [15,21,22].

#### 1.1.2. Chile

The pandemic came at a time of strong discontent and social protest in response to a neoliberal model inherited from the Pinochet dictatorship. Public policies implemented by the Chilean government focused on social distancing and population control. A State of catastrophe was implemented, which included a curfew. Subsequently, the use of masks was made mandatory; sanitary customs were created; partial and total quarantines were established, differentiated by cities, with fines and sanctions for those who violated the lockdown. According to the Chilean neoliberal model, the state plays a subsidiary role, reflected in public economic policies, which initially focused on “employment protection”, giving the possibility of suspending workers’ salaries and tax payments for 3 months in favor of companies [15,21,23].

#### 1.1.3. Ecuador

In this country, given the situation of a public health system in decline, in the first months, a first lockdown and curfew were decreed as a contagion control measure, allowing the sectors of primary need to maintain mobility to fulfill their functions; in addition, the country closed its borders, educational institutions and shopping centers were closed, and the use of masks in public spaces was established. Subsequently, the country’s sanitary traffic light was planned, with different measures according to the number of infections. At the end of May 2020, “social distancing” was proposed, a period in which the different municipalities would develop pilot plans for the progressive return to operation of the productive sectors, including educational centers and recreational services such as shopping centers [15,21,24].

#### 1.1.4. Mexico

At the beginning of the pandemic, the national promotion of basic hygiene measures was initiated with emphasis on the most vulnerable populations in terms of health status. In view of the increase in infections and community transmission of the virus, preventive measures were intensified. Classes, events, and meetings were suspended, as well as all actions involving crowds of people. Although lockdown was indispensable, it was never made mandatory, since more than 50% of the population was in the informal economy; hence, they lacked social security, and it was impossible for this population to protect themselves. In the case of the formal economy, people worked exclusively in sectors considered basic necessities, such as services and basic products [15,21,25].

#### 1.1.5. Peru

The government, before other countries in the region, declared a national health emergency and issued various measures for the prevention and control of the disease. At the same time, it ordered social isolation or so-called compulsory lockdown throughout the country. A national quarantine extended to all Peruvian citizens was indicated from 16 March to 31 December 2020, but with restrictive measures in a differentiated manner for each department or province, according to the prevalence of positive cases from 1 July to 31 December 2020. In 2021, some economic activities were reestablished to avoid further poverty and control the economic impact for the country, with some restrictions in the province and limitations on the exercise of freedom of movement of people [15,21,26].

#### 1.1.6. Spain

A total lockdown of the population was declared by means of a state of alarm. The declaration of the state of alarm implied the closure of establishments and workplaces, as well as the prohibition of certain activities. Freedom of movement was restricted with the exception of the purchase of basic necessities, the attendance of health services, and the assistance and care of dependents. By the end of April 2020, COVID-19 cases were declining, and a four-phase de-escalation or closure plan called the “plan for transition to a new normal” was approved. The plan included a series of measures, while easing restrictions on mobility and social contact, and allowing certain businesses and services to open to the public. After the de-escalation, decisions on control measures began to fall to the Spanish autonomous communities [21,27].

### 1.2. Mental Health and Self-Perceived Health Problems

A review quantified the prevalence and burden of depressive and anxiety disorders by age, sex, and location worldwide. Women and younger age groups were more affected than men and older age groups. They estimated large increases in prevalence in Latin America and the Caribbean, despite not finding any surveys from these regions that met their inclusion criteria [3].

In studies of countries in the region, in addition to the factors mentioned (i.e., considering women and younger people), worsening mental health is associated with being attentive to news about the pandemic, having someone close diagnosed with COVID-19, the possibility of getting sick, loss of contact with peers [28], feeling a greater burden in taking care of children, taking medication on a regular basis, having a lower family income [29], not having a partner [6], and having poor sleep quality [30]. In the case of self-perceived health, there are studies that showed a relationship with having informal work, being a student or retired, reporting gender violence [14], having solely public healthcare system access, having COVID-19, and presence of any chronic illness [24], factors that increased the probability of having poor self-reported health status. However, there is limited evidence in comparative studies about mental health and self-perceived health in the region.

### 1.3. Research Question and Objective

On the basis of this diverse context in terms of the measures implemented, we posed the following questions: (a) How has mental health and self-perceived health been according to the measures implemented for the management of the pandemic by COVID-19 in Latin American countries and Spain? (b) How did these outcomes differ according to inequality axes? Therefore, the aim of this study was to compare mental health status and self-perceived health in several LA countries and Spain during the lockdown of the first wave of the COVID-19 pandemic, according to several social factors.

## 2. Materials and Methods

### 2.1. Study Design and Data Source

A cross-sectional descriptive study was conducted using a self-administered online survey of people aged 18 and over living in Brazil, Chile, Ecuador, Mexico, Peru, and Spain. Data collection was carried out in 2020 during the first wave between June and August (Brazil), May and August (Chile and Mexico), July and October (Ecuador), July and September (Peru), and April and May (Spain).

The questionnaire was designed by a multidisciplinary research team in Spain and adapted to the specific context of each country. A pilot study was conducted prior to dissemination in order to represent the sociodemographic diversity of each country’s population. At the beginning of the survey, the objective of the study and the duration of the survey were explained, which lasted approximately 10 min, including the reading and signing of the informed consent. In Spain, the REDCap (Research Electronic Data Capture) platform was used, an electronic data capture tool hosted at the Fundació Institut Universitari per a la recerca a l’Atenció Primària de Salut Jordi Gol i Gurina (IDIAPJGol). REDCap is a secure, web-based software platform designed to support data capture for research studies, providing (1) an intuitive interface for validated data capture, (2) audit trails for tracking data manipulation and export procedures, (3) automated export procedures for seamless data downloads to common statistical packages, and (4) procedures for data integration and interoperability with external sources [31,32]. For LA, survey data were collected and managed using SurveyMonkey^®^ electronic data capture tools (hosted by IDIAPJGol). Our study was approved by the Research Ethics Committee of the Institut de Recerca en Atenció Primària Jordi Gol i Gurina (IDIAPJGol) (ref. REC 20/063-PCV).

### 2.2. Sampling

Data collection was carried out through the online platforms of each of the participating centers in the different countries and their respective social networks and mass media, using convenience and snowball sampling techniques.

### 2.3. Variables

The main study variables were mental health problems (anxiety and depression) and SPH. Anxiety was defined as persistent worry and anticipatory responses to future threats, as measured by the Generalized Anxiety Disorder (GAD-7) screening tool; it was classified according to the score obtained as “normal/no anxiety” and “moderate to severe” [33,34]. Depression was defined as marked feelings of sadness, emptiness, or irritability, assessed by the Patient Health Questionnaire (PHQ-9) and classified as “minimal/mild” or “moderate/severe” [33,34]. The SPH (“How would you say your overall health is?”) has five Likert-scale response options, which were recategorized into “good” and “poor”. For the mental health outcomes, depression and anxiety scales were used, which have been validated in all countries participating in the study [35,36,37,38,39,40].

The sociodemographic variables and those related to pandemic lockdown were gender identity, educational level, age, indigenous group membership, country of birth, pre-pandemic employment status, change in employment status, housing tenure, perception of adequate housing, household composition, presence of children and/or dependents in the household, household work, concern about living with household members, and concern about school education (see Table A1 for details of variables).

### 2.4. Statistical Analysis

A descriptive analysis of all variables of interest was performed to obtain absolute numbers and percentages. The chi-square test was used to determine if there were differences between sexes. The prevalence of self-perceived poor health, anxiety, and depression was calculated. Analyses were stratified by sex (men/women) and country. All statistical analyses were performed using Stata version 15.1 software.

## 3. Results

Data from 39,006 people who completed the survey were analyzed (see Table A2 for distribution of missing values). Of the total number of respondents, the majority were female (71.9%), between 35 and 64 years old (58.8%), with a university education (73.0%), a trend that was repeated in all countries in the following order of representation: Brazil (35.8%), Mexico (21.5%), Spain (18.7%), Chile (14.4%), Ecuador (6.8%), and Peru (2.9%) (Table 1) (see Table A3 for complete data).

### 3.1. Self-Perceived Health (SPH)

There was a higher prevalence of poor SPH in women than in men in all countries studied. The highest prevalence of poor SPH was found in Peru (men = 26%; women = 34%) and Brazil (men = 21%; women = 25%), while the lowest was found in Spain (men = 9%; women = 12%) and Ecuador (men = 12%; women = 18%). There was a higher prevalence of poor SPH among those who reported belonging to indigenous groups, except for indigenous women in Peru and indigenous men in Chile (Table 2). In Chile, Mexico, and Spain, SPH worsened with increased age, while no gradient was observed in the other countries (Figure 1a).

In most cases there was a higher prevalence of poor SPH among those who were not employed, and it was higher in women in all countries (except Peru). Among those in paid employment, poor SPH was related to worsening employment status, especially among women in Peru. For home tenure, there was a gradient in the prevalence of poorer SPH across countries, with a lower prevalence among those living in their own home, a higher prevalence among those living in rented houses, and an even higher prevalence among those living in someone else’s home. The highest prevalence was reported among women living in someone else’s home in Peru (45%) and Brazil (38%) (Table 2).

Regarding family composition and unpaid care work, there were no clear trends between the number of household members and the presence of children. However, there was a higher prevalence of poorer SPH in those living with dependents in all countries for both genders (except men in Ecuador), being higher in women in Peru (40%), Brazil (27%), and Mexico (27%). In addition, there was a higher prevalence of poor SPH among those who performed most of the housework in the home, especially among women in Peru (45%). Regarding concern for household members and their school education, there was a higher prevalence of poor SPH for those who reported worrying a great deal or a lot in all countries studied, being higher in women from Peru (39%) and Brazil (27%) (Table 2).

### 3.2. Mental Health: Anxiety and Depression

The highest prevalence of mental health problems was found in Chile (anxiety: men = 43%, women = 58%; depression: men = 31%, women = 42%) and Ecuador (anxiety: men = 35%, women = 46%; depression: men = 32%, women = 43%), and the lowest prevalence was found in Spain (anxiety: men = 18%, women = 31%; depression: men = 16%, women = 28%). There was a higher prevalence of anxiety and depressive symptoms in women regardless of country and sociodemographic characteristics. The highest prevalence of anxiety and depression was observed in the younger groups, decreasing in the older groups in all countries (Figure 1b,c). A higher prevalence of anxiety and depression was observed in women belonging to indigenous groups (Table 3 and Table 4).

The prevalence of depression was higher among those who were unemployed prior to the pandemic, while it was variable for anxiety; however, it was higher in those who reported that their employment status worsened during the pandemic (for anxiety and depression), standing out women in Chile (anxiety 63%; depression 48%). The prevalence of mental health problems) increased for those who lived in rented houses and those who lived in someone else’s home. Likewise, those who considered that their housing was not adequate had a higher prevalence of mental health problems; this tendency was greater in women than in men, especially in Chile (anxiety: men = 55%, women = 72%; depression: men = 45%, women = 62%) and Ecuador (anxiety: men = 48%, women = 63%; depression men = 43%, women = 54%) (Table 3 and Table 4).

Regarding living together during lockdown, no clear trends were observed between the number of household members and the prevalence of anxiety and depression. Living with children was associated with higher anxiety for women in all countries, and living with dependents led to a higher prevalence of anxiety and depression. These results are consistent with the burden of care and concern about living with household members and school education for all women, with higher prevalence among women in Brazil (depression) and Chile (anxiety) (Table 3 and Table 4).

## 4. Discussion

This study showed the results of the prevalence of anxiety, depression, and SPH in Brazil, Chile, Ecuador, Peru, Mexico, and Spain during the lockdown in the first wave of the COVID-19 pandemic. Our findings highlight that there was a higher prevalence of poor SPH, especially in Peru, and a higher impact on mental health in Chile and Ecuador. Women were the most affected in all the countries studied. We observed an age gradient; younger persons had a higher presence of symptoms of anxiety and depression, but not poor SPH. Our results also suggest that there were social determinants related to a higher prevalence of poor SPH and mental health problems, especially in women, such as pre-pandemic unemployment, worse working conditions, the perception of inadequate housing, and a higher burden of unpaid care work.

We observed differences in overall prevalence in our study in LA countries compared to Spain. The best results in mental health measures and SPH in Spain may indicate the relationship between the social and material circumstances in which people lived prior to the pandemic, as well as governance and its relationship with the impact on mental health. Many LA countries announced emergency fiscal plans with direct cash transfer programs to the most impoverished households, but maintaining mobility restrictions, with the subsequent loss of (mostly informal) employment and reduction in labor income, thus increasing structural inequality gaps [41]. On the other hand, in Spain, the measures were based on social welfare policies and a shorter lockdown duration, which could partly explain the lower prevalence of poor mental and SPH compared to LA countries. In this line, it is essential to consolidate universal social protection systems in LA, including social security, education, and health, which are relevant for social welfare, the effective enjoyment of rights, and the improvement of population’s health [42], especially in times of crisis and uncertainty. Moreover, it is necessary to add social sciences and women in management to be sensitive to the importance of social as continuous change, social reproduction, and gender inequalities [43,44]

This study found a higher prevalence of poor SPH and mental health in all study countries among those who were unemployed, as well as among those who were working but whose employment situation worsened, as in the case for women in Peru. This result is consistent with the increase in informal work and economic fluctuations in this country [45]. In the LA context, those unemployed during the pandemic reported more stress than those employed [46]. LA and the Caribbean are the regions with the greatest impact on formal employment worldwide [17]; thus, the impact of the pandemic was related to concern about the lack of availability of material resources [46], causing uncertainty in people and damaging their mental health [47]. In addition, both Chile, immersed in a political and institutional crisis, due to the strong discontent and social protest in response to a neoliberal model inherited from the Pinochet dictatorship [14,48], and Ecuador, which was experiencing severe economic and governmental management problems [15], had the worst mental health outcomes. Previous studies showed an association between suffering from mental health problems and living in historical contexts characterized by a lack of freedom and unstable environments [49].

On the other hand, lockdown has made the characteristics of housing and tenure relevant factors in responding to the demands of control measures. Poor SPH and mental health were lower in those who lived in their own home, while it worsened among those who live in rented houses and was further aggravated among those who lived in someone else’s home. This situation was seen in other studies, which described housing as a factor that produces stress and anxiety, when it is of poor quality, small, or perceived as inadequate to house the inhabitants of the household [50]. On the other hand, the size of the dwelling played an essential role, since it was shown that being confined to larger spaces favors SPH [14]. Another aspect not considered in these types of restrictions are the potential effects due to the energy poverty existing in the region [51], the obligation to stay in a place without good conditions (in the southern cone, autumn began on that date), and the potential effects on people’s health [52,53]. This suggests that homogeneous pandemic containment strategies fracture society and deepen existing vulnerabilities [43].

Younger people had the highest prevalence of anxiety and depression, especially in women, decreasing in older people in all countries, except for men in Peru. These results are consistent with other studies that showed a decrease in the occurrence of mental health problems with increasing age [54], despite the fact that COVID-19 threatens the physical health of the older population, related to social issues [43]. In LA, educational centers were closed for an average of more than 1 year, and, despite the boost of virtual classes, this situation increased the effects of the digital divide and emotional apathy [20]. It also increased uncertainty about daily life, as well as its financial burdens, and the continuity of learning [20], causing discomfort due to the absence of face-to-face interaction with teachers and mates [14]. On the other hand, other studies associated poor mental health in adolescents and young adults with low expectations of being able to finish their studies and the uncertainty of entering the productive world [20]. Likewise, the worst mental health outcomes in women could be explained by the negative impact of educational trajectories when there are sociopolitical and economic crises that deepen gender inequalities [55]. On the other hand, women reported that they were exhausted by having to combine caregiving, teleworking, and emotional support, with no possibility of recovery [56].

Those concerned about living with family members and school education showed a higher prevalence of poor mental health problems. This situation is framed by the crisis of care, which refers to the challenges faced by neoliberal societies to ensure social reproduction, including caring for oneself and others, the time spent maintaining physical spaces, the organization of the necessary resources, and human reproduction [57]. In addition to the activity of caring itself, assuming organizational responsibilities in times of compulsory cohabitation reinforces the need to recognize and redistribute care work [58]. For this reason, it is imperative to establish state policies that favor co-responsibility between members of the family and social sphere, overcoming gender stereotypes. This allows us to recognize the importance of care and domestic work for the economic reproduction and wellbeing of society as one of the ways to overcome the feminization of poverty [59].

### Limitations and Strengths

All surveys were conducted through online tools, excluding people without access to technology and the survey itself. This may have led to an overrepresentation of responses from people with higher levels of education [60]. However, given the health context at the time of the study, this was the most convenient way to obtain the information and brings us closer to the important inequalities that exist. Another limitation was the difference in the size of the samples collected in each country. Therefore, the results should be interpreted with caution, since the reported prevalence was not population-based, but rather referred to the social groups in our study. Among the strengths, this study is one of the first to explore the effects of lockdown in different LA countries and Spain, allowing us to have a global picture of what happened during the first wave of the pandemic, through the stratification of many sociodemographic characteristics and health outcomes. This implies considering mental health from a situated and contextual perspective, in which the strength of the state and the capacity of individuals and support networks to respond to crises are especially relevant. In future research, it would be interesting to have longitudinal studies and qualitative studies to follow the impact of the pandemic on mental health and self-perceived health over time in the region.

## 5. Conclusions

In Latin America and Spain, the social and health crisis generated by the first wave of COVID-19 has not affected all countries and social groups equally. The impact of lockdown has particularly affected women and young people. Chile and Ecuador had the worst mental health outcomes, Peru had the worst SPH, and Spain had better results, mainly related to the difference in lockdown characteristics, the social context, and socioeconomic factors, especially those related to income (i.e., employment, work condition, and perception of adequate housing). The lack of preparedness and the adoption of a reactive approach underlie many mistakes in handling the COVID-19 pandemic. We need a vision with a proactive approach to planetary health prevention, which is suited for addressing the neglected systemic determinants of health that generate disease, inequality, and environmental degradation. This implies including different actors and expertise to understand the health and social crisis from a holistic point of view. This highlights, among structural determinants (such as housing conditions), the importance of conditions of social reproduction and the provision of mental health treatment (specialized public mental health service). As suggested by this study, there is an urgent need today to promote community resilience strategies, with policies and interventions that protect the mental health of the population in emergencies such as COVID-19.

## Figures and Tables

**Figure 1 ijerph-20-05722-f001:**
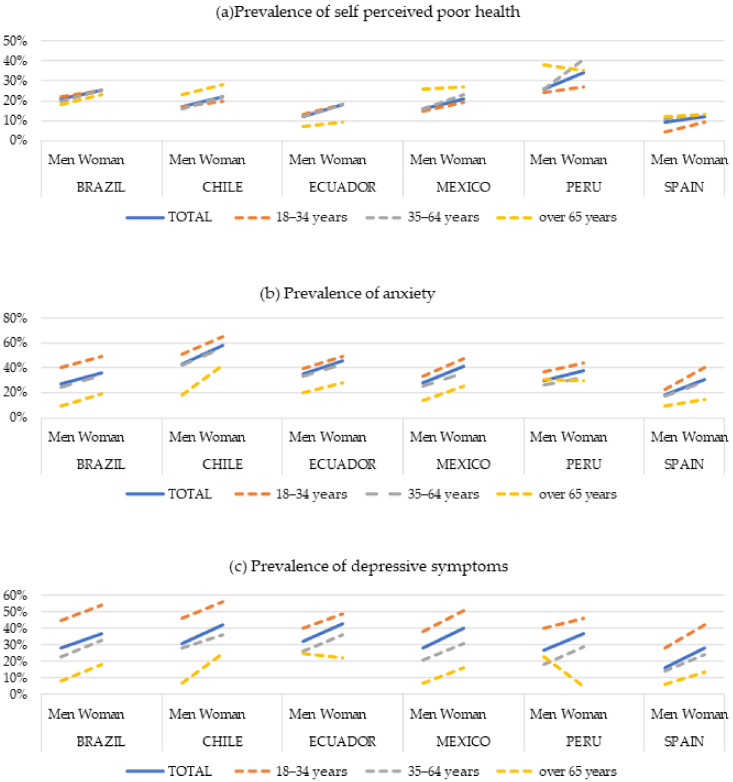
Prevalence of (**a**) self-perceived poor health, (**b**) anxiety, and (**c**) depressive symptoms according to age and gender in Brazil, Chile, Ecuador, Mexico, Peru and Spain in the first wave of COVID-19 lockdown.

**Table 1 ijerph-20-05722-t001:** Sociodemographic characteristics and variables related to social factors of the participants by sex in Brazil, Chile, Ecuador, Mexico, Peru, and Spain during the first wave of COVID-19 lockdown.

		Brazil			Chile				Ecuador			México			Peru			Spain		
		n = 13,943 (35.75%)			n = 5612 (14.39%)				n = 2653 (6.8%)			n = 8396 (21.52%)			n = 1122 (2.88%)			n = 7280 (18.66%)		
		Men	Woman		Men	Woman		Men	Woman		Men	Woman		Men	Woman		Men	Woman	
		n = 2917	n = 11,026		n = 2095	n = 3517		n = 853	n = 1800		n = 2648	n = 5748		n = 359	n = 763		n = 2101	n = 5179	
		% M	% W	*p*	% M	% W	*p*	% M	% W	*p*	% M	% W	*p*	% M	% W	*p*	% M	% W	*p*
Educational level	Basic or bachelor’s degree	21.8	18.6	**	15.5	14.4		11.0	13.0		5.1	4.2	**	12.4	16.2		21.0	17.3	**
	Technical studies	8.1	5.7		16.0	15.4		3.8	3.6		23.2	27.7		14.2	16.6		10.8	10.5	
	University studies	70.1	75.7		68.6	70.2		85.3	83.5		71.7	68.1		73.5	67.2		68.2	72.2	
Age	18–34	32.3	27.7	**	28.1	31.7	*	41.7	55.2	**	44.7	48.1	**	39.3	50.6	**	23.5	27.0	**
	35–64	57.5	63.7		65.1	62.3		54.9	43.6		50.4	48.7		56.3	46.1		64.0	66.2	
	≥65	10.2	8.7		6.8	6.1		3.4	1.2		5.0	3.2		4.5	3.3		12.5	6.8	
Indigenous group membership ^1^	No	66.1	68.5	*	89.3	90.4		90.3	93.0	*	89.5	90.0		83.3	85.8				
	Yes	33.9	31.5		10.8	9.6		9.7	7.0		10.5	10.0		16.7	14.2				
Country of origin	Born in country of study	97.6	98.2	*	91.8	92.2		94.8	96.3		97.9	98.4		96.7	96.5		90.8	90.7	
	Migrant	2.4	1.8		8.2	7.8		5.2	3.7		2.1	1.7		3.3	3.5		9.2	9.3	
Pre-pandemic employment status	Employed	66.5	56.9	**	88.1	78.8	**	82.3	73.8	**	70.2	62.7	**	83.2	68.9	**	71.9	70.4	
	Unemployed	33.5	43.1		11.9	21.2		17.7	26.2		29.8	37.3		16.8	31.1		28.2	29.6	
Change in employment status during the pandemic	No change/improvement in employment condition	47.7	49.7		41.5	44.3	*	30.2	28.2		49.6	47.8		48.3	47.1		59.6	59.3	
	Worsening in employment condition	52.3	50.3		58.5	55.7		69.8	71.8		50.4	52.3		51.8	52.9		40.4	40.7	
Housing tenure	Own home	67.3	68.7	**	58.1	57.7		64.2	65.0		63.0	61.5	**	59.1	61.2		69.6	71.8	
	Lease or rent	21.1	23.7		32.1	30.8		28.4	26.5		24.5	21.7		29.9	26.2		26.4	25.0	
	Living in someone else’s home	5.6	6.7		9.8	11.5		7.5	8.5		12.5	16.8		11.0	12.6		4.0	3.3	
Perception of adequate housing	Suitable for confinement	81.7	81.8		84.9	87.7	0	86.8	86.4		79.4	77.2	*	81.9	75.8	*	89.9	90.0	
	Not suitable for confinement	18.3	18.2		15.1	12.3		13.2	13.6		20.6	22.8		18.2	24.3		10.1	10.0	
Household composition	Living alone	19.0	17.4		12.5	11.3		11.8	8.6	*	11.6	9.1	**	8.6	6.7		19.3	17.7	
	Living with other people	81.0	82.6		87.5	88.7		88.2	91.4		88.4	90.9		91.4	93.3		80.7	82.3	
Presence of minors in the household	No	67.8	62.5	**	56.4	55.1		50.1	47.9		53.8	48.4	**	42.0	37.3		70.0	64.3	**
	Yes	32.2	37.5		43.6	44.9		49.9	52.1		46.2	51.6		58.0	62.7		30.0	35.7	
Presence of dependents in the household	No	43.0	43.2		64.8	59.8	**	58.4	48.6	**	62.7	51.9	**	48.0	43.4		87.9	84.8	**
	Yes	57.1	56.8		35.2	40.2		41.6	51.4		37.4	48.1		52.0	56.6		12.1	15.2	
Household work	Other persons/equally among household members	84.0	48.4	**	87.5	55.2	**	89.1	65.0	**	88.8	66.3	**	90.8	70.0	**			
	Mostly by myself	15.9	51.6		12.5	44.8		10.9	35.0		11.2	33.7		9.3	30.0				
Concern about living with household members	Nothing or little	30.9	32.2		46.2	51.3	**	42.8	44.7		48.0	49.9		36.2	37.0		79.6	76.2	*
	Moderate, quite a bit, or a lot	69.1	67.8		53.3	48.8		57.2	55.3		52.0	50.1		63.8	63.0		20.4	23.9	
Concern about schooling	Nothing or little	17.6	22.0	**	32.6	41.1	**	13.9	13.0		26.6	27.0		24.9	26.3		43.6	51.5	**
	Moderate, quite a bit, or a lot	82.4	78.0		67.4	48.9		86.1	87.0		73.4	73.0		75.1	73.8		56.4	48.5	

% M = percentage of men; % W = percentage of women. ^1^ Race/skin color self-declaration in case of Brazil (no: white; yes: black, brown, yellow, or indigenous). *p* = statistical significance between men and women derived from the chi-squared test; ** *p* < 0.001, * *p* < 0.05. NA: not applicable (no information on this variable).

**Table 2 ijerph-20-05722-t002:** Prevalence of self-perceived poor health according to gender and sociodemographic characteristics in Brazil, Chile, Ecuador, Mexico, Peru, and Spain during the first wave of COVID-19 lockdown.

		Brasil		Chile		Ecuador		Mexico		Perú		Spain	
		Men	Women	Men	Women	Men	Women	Men	Women	Men	Women	Men	Women
		N = 2526	N = 9739	N = 1928	N = 3288	N = 759	N = 1637	N = 2405	N = 5320	N = 323	N = 688	N = 2092	N= 5162
		Prevalence (IC95%)	Prevalence (IC95%)	Prevalence (IC95%)	Prevalence (IC95%)	Prevalence (IC95%)	Prevalence (IC95%)	Prevalence (IC95%)	Prevalence (IC95%)	Prevalence (IC95%)	Prevalence (IC95%)	Prevalence (IC95%)	Prevalence (IC95%)
Self-perceived health		21 (19–22)	25 (24–25)	17 (15–18)	22 (20–23)	12 (10–15)	18 (16–19)	16 (14–17)	21 (20–22)	26 (21–31)	34 (31–38)	9 (8–11)	12 (11–13)
Educational level	Basic or bachelor’s degree	29 (25–33)	37 (35–39)	24 (19–28)	34 (30–38)	20 (11–28)	21 (15–26)	24 (16–31)	34 (28–40)	35 (20–50)	36 (27–45)	12 (9–15)	20 (18–23)
	Technical studies	25 (19–31)	35 (31–39)	23 (18–27)	29 (25–33)	07 (0–17)	30 (18–42)	19 (16–23)	26 (23–28)	22 (9–35)	32 (23–40)	12 (7–16)	16 (12–19)
	University studies	18 (16–20)	21 (20–22)	14 (12–15)	18 (16–19)	11 (9–14)	16 (15–18)	14 (13–16)	18 (17–20)	26 (20–31)	34 (30–38)	8 (7–10)	10 (9–11)
Age	18–34	22 (19–25)	25 (24–27)	17 (13–20)	20 (18–23)	13 (9–17)	18 (15–20)	15 (13–17)	19 (17–20)	24 (16–32)	27 (23–32)	4 (2–6)	9 (7–10)
	35–64	20 (18–22)	25 (23–26)	16 (14–18)	22 (20–23)	12 (9–15)	18 (15–20)	16 (14–18)	23 (21–25)	26 (20–33)	41 (36–47)	11 (9–12)	13 (12–15)
	≥65	18 (14–23)	23 (21–26)	23 (16–30)	28 (22–35)	07 (0–17)	09 (0–22)	26 (18–34)	27 (20–33)	38 (12–65)	35 (15–54)	12 (08–16)	13 (10–17)
Indigenous group membership ^1^	No	19 (17–21)	22 (21–23)	17 (15–19)	21 (20–23)	11 (9–13)	16 (14–18)	15 (14–17)	21 (20–22)	23 (18–28)	34 (30–38)	NA	NA
	Yes	23 (20–26)	31 (29–33)	16 (11–21)	27 (22–32)	25 (15–35)	36 (27–44)	19 (14–24)	23 (20–27)	40 (27–54)	33 (24–43)	NA	NA
Country of origin	Born in the country of study	21 (19–22)	25 (24–26)	17 (15–19)	22 (21–24)	12 (9–14)	18 (16–19)	16 (14–17)	21 (20–22)	NA	NA	9 (7–10)	12 (11–13)
	Migrant	24 (13–35)	23 (17–29)	13 (06–19)	13 (08–18)	21 (8–33)	17 (8–27)	10 (2–18)	14 (7–21)	NA	NA	14 (10–19)	15 (12–18)
Pre-pandemic employment status,	Employed	19 (17–21)	22 (21–24)	15 (14–17)	19 (17–20)	11 (9–14)	17 (15–19)	15 (14–17)	21 (19–22)	26 (20–31)	35 (30–39)	8 (7–9)	9 (8–10)
	Unemployed	24 (21–27)	26 (25–28)	23 (18–29)	30 (26–33)	16 (9–22)	19 (15–23)	17 (14–19)	22 (20–24)	27 (15–39)	33 (26–39)	13 (10–16)	19 (17–21)
Change in employment status during the pandemic	No change/improvement in employment condition	18 (15–20)	22 (21–23)	13 (11–15)	18 (16–20)	9 (5–13)	15 (12–18)	13 (12–15)	18 (17–20)	18 (12–25)	29 (24–35)	10 (8–12)	12 (10–13)
	Worsening in employ–ment condition	23 (21–26)	28 (26–29)	19 (17–22)	24 (23–26)	13 (11–16)	19 (16–21)	18 (16–20)	24 (22–25)	35 (27–42)	38 (33–44)	9 (7–11)	13 (12–15)
Housing tenure	Own home	19 (17–21)	23 (22–24)	16 (13–18)	19 (17–21)	10 (7–13)	16 (14–18)	15 (14–17)	19 (18–21)	25 (19–31)	33 (28–37)	10 (8–12)	12 (11–13)
	Lease or rent	22 (19–25)	27 (25–29)	17 (14–19)	23 (21–26)	17 (12–22)	20 (16–24)	15 (12–18)	21 (18–23)	27 (18–36)	31 (24–38)	7 (5–10)	11 (9–13)
	Living in someone else’s home	32 (24–40)	38 (34–41)	24 (18–31)	29 (25–34)	14 (5–23)	21 (15–28)	19 (15–24)	28 (25–31)	31 (16–46)	45 (35–56)	08 (2–14)	19 (13–24)
Perception of adequate housing	Suitable for confinement	18 (16–20)	22 (21–23)	15 (13–17)	19 (18–21)	10 (8–12)	16 (14–18)	15 (13–16)	19 (17–20)	23 (18–28)	32 (28–36)	9 (8–10)	11 (10–12)
	Not suitable for confinement	31 (27–35)	37 (35–40)	27 (22–32)	38 (33–43)	26 (18–35)	29 (23–35)	20 (17–24)	30 (27–33)	42 (29–55)	42 (34–49)	14 (9–18)	20 (16–23)
Household composition	Living alone	20 (17–24)	23 (21–25)	15 (11–20)	18 (14–22)	14 (7–21)	15 (9–21)	17 (12–21)	17 (14–20)	14 (01–26)	22 (10–34)	10 (07–13)	12 (10–14)
	Living with other people	21 (19–22)	25 (24–26)	17 (15–19)	22 (21–24)	12 (10–14)	18 (16–20)	16 (14–17)	22 (20–23)	27 (22–32)	35 (31–39)	9 (8–10)	12 (11–13)
Presence of minors in the household,	No	20 (18–22)	24 (23–25)	18 (15–20)	22 (20–25)	12 (9–16)	15 (12–17)	16 (13–18)	20 (18–21)	31 (22–40)	32 (25–38)	10 (6–13)	15 (12–17)
	Yes	21 (19–24)	26 (24–27)	16 (14–19)	22 (20–24)	12 (8–15)	20 (18–23)	16 (14–18)	23 (21–25)	25 (19–31)	37 (32–41)	6 (3–08)	9 (7–11)
Presence of dependents in the household	No	17 (14–20)	21 (19–22)	13 (11–15)	19 (17–21)	12 (8–15)	12 (10–15)	12 (10–13)	16 (14–17)	22 (15–30)	27 (21–32)	17 (11–23)	16 (12–19)
	Yes	22 (20–24)	27 (26–28)	23 (20–26)	26 (24–29)	12 (9–16)	22 (19–25)	21 (19–24)	27 (25–28)	31 (24–38)	40 (35–45)	9 (4–15)	13 (9–17)
Household work	Other persons/equally among household members	20 (18–22)	23 (22–25)	17 (15–19)	20 (18–22)	12 (10–15)	16 (14–18)	15 (14–17)	20 (18–21)	27 (22–32)	31 (27–35)	NA	NA
	Mostly by myself	24 (19–29)	26 (25–28)	18 (13–24)	24 (22–26)	11 (4–18)	21 (18–25)	19 (14–24)	25 (23–27)	30 (12–47)	45 (38–52)	NA	NA
Concern about living with household members	Nothing or little	14 (12–17)	19 (18–21)	13 (11–15)	19 (17–21)	9 (6–13)	13 (10–16)	13 (11–15)	19 (17–20)	19 (12–27)	26 (21–32)	8 (7–10)	10 (9–11)
	Moderate, quite a bit or a lot	23 (21–26)	28 (26–29)	20 (18–23)	25 (23–28)	14 (10–17)	22 (19–24)	18 (16–20)	24 (23–26)	32 (25–39)	40 (35–45)	13 (9–16)	18 (16–21)
Concern about schooling	Nothing or little	26 (19–33)	22 (19–25)	17 (13–22)	20 (16–23)	11 (3–20)	14 (8–21)	16 (12–20)	21 (18–24)	14 (03–24)	30 (21–38)	6 (3–9)	10 (8–11)
	Moderate, quite a bit or a lot	21 (18–24)	27 (25–29)	16 (13–19)	23 (20–26)	12 (8–15)	21 (18–24)	16 (13–18)	24 (22–26)	27 (20–35)	39 (34–45)	9 (6–11)	14 (12–16)

NA: not applicable (no information on this variable). ^1^ Race/skin color self-declaration in case of Brazil (no: white; yes: black, brown, yellow, or indigenous).

**Table 3 ijerph-20-05722-t003:** Prevalence of anxiety according to gender and sociodemographic characteristics in Brazil, Chile, Ecuador, Mexico, Peru, and Spain during the first wave of COVID-19 lockdown.

		Brasil		Chile		Ecuador		Mexico		Perú		Spain	
		Men	Women	Men	Women	Men	Women	Men	Women	Men	Women	Men	Women
		N = 2404	N = 9307	N = 1809	N = 3167	N = 711	N = 1496	N = 2210	N = 4867	N = 299	N = 645	N = 2077	N = 5155
		Prevalence (IC95%)	Prevalence (IC95%)	Prevalence (IC95%)	Prevalence (IC95%)	Prevalence (IC95%)	Prevalence (IC95%)	Prevalence (IC95%)	Prevalence (IC95%)	Prevalence (IC95%)	Prevalence (IC95%)	Prevalence (IC95%)	Prevalence (IC95%)
Anxiety		27 (25–29)	37 (36–38)	43 (41–45)	58 (56–60)	35 (32–39)	46 (43–48)	28 (26–30)	41 (40–43)	30 (25–36)	38 (34–42)	18 (16–19)	31 (30–33)
Educational level	Basic or bachelor’s degree	33 (29–37)	46 (44–48)	51 (46–57)	69 (64–73)	44 (32–55)	48 (40–55)	31 (21–40)	42 (35–50)	40 (24–56)	52 (43–62)	18 (15–22)	38 (35–41)
	Technical studies	35 (28–41)	47 (43–51)	44 (39–50)	63 (58–67)	32 (14–50)	45 (31–59)	30 (26–34)	44 (41–46)	32 (17–46)	41 (31–50)	21 (15–26)	35 (31–39)
	University studies	24 (22–26)	34 (33–35)	41 (38–44)	55 (52–57)	34 (30–38)	46 (43–48)	28 (25–30)	40 (39–42)	29 (23–35)	34 (30–39)	17 (15–19)	29 (28–31)
Age	18–34	40 (36–43)	49 (47–51)	51 (47–56)	65 (62–68)	39 (34–45)	49 (45–52)	33 (31–36)	47 (45–49)	37 (28–46)	44 (39–50)	23 (20–27)	40 (37–43)
	35–64	24 (22–26)	34 (33–35)	42 (39–45)	56 (53–58)	33 (29–38)	43 (39–46)	25 (22–27)	36 (34–38)	26 (20–33)	32 (27–38)	17 (15–19)	29 (28–31)
	≥65	9 (6–13)	19 (16–21)	18 (11–24)	42 (35–49)	20 (4–36)	28 (7–48)	14 (7–21)	25 (18–32)	31 (6–56)	30 (10–50)	9 (6–13)	15 (12–19)
Indigenous group membership ^1^	No	27 (24–29)	36 (35–37)	43 (41–46)	57 (55–59)	35 (32–39)	45 (42–47)	29 (27–31)	41 (40–42)	28 (22–33)	38 (34–42)	NA	NA
	Yes	28 (25–31)	39 (37–40)	43 (36–50)	65 (59–70)	35 (23–47)	57 (48–67)	22 (17–28)	44 (39–48)	45 (30–59)	38 (27–48)	NA	NA
Country of origin	Born in the country of study	27 (26–29)	37 (36–38)	44 (41–46)	58 (56–60)	36 (32–39)	46 (43–48)	28 (26–30)	41 (40–43)	NA	NA	17 (15–19)	31 (30–33)
	Migrant	12 (04–21)	30 (23–38)	33 (23–42)	53 (45–61)	29 (14–44)	47 (34–60)	21 (09–32)	42 (32–53)	NA	NA	22 (17–27)	33 (29–36)
Pre-pandemic employment status,	Employed	27 (25–29)	37 (36–39)	43 (41–46)	57 (55–59)	35 (31–39)	45 (42–48)	28 (26–30)	41 (40–43)	29 (23–34)	36 (32–41)	19 (17–21)	31 (29–32)
	Unemployed	26 (23–30)	36 (34–38)	39 (32–45)	61 (57–65)	39 (30–47)	47 (42–53)	28 (25–32)	41 (39–43)	40 (26–54)	44 (37–51)	16 (13–18)	32 (30–35)
Change in employment status during the pandemic	No change/improvement in employment condition	17 (15–20)	29 (28–31)	35 (32–39)	52 (49–54)	24 (18–30)	36 (31–41)	21 (19–24)	34 (32–36)	22 (15–28)	32 (26–37)	15 (13–17)	27 (25–28)
	Worsening in employment condition	36 (33–38)	44 (42–45)	49 (46–52)	63 (61–65)	40 (36–44)	50 (47–53)	35 (32–38)	48 (46–50)	39 (31–47)	44 (38–50)	22 (19–25)	38 (36–40)
Housing tenure	Own home	24 (22–26)	34 (33–35)	39 (36–42)	54 (51–56)	34 (30–38)	43 (40–46)	26 (24–28)	38 (36–40)	25 (19–32)	36 (31–40)	16 (14–18)	30 (29–32)
	Lease or rent	31 (27–34)	42 (40–44)	48 (44–52)	63 (60–66)	37 (30–44)	50 (45–55)	33 (29–37)	47 (44–50)	37 (27–47)	44 (36–51)	20 (16–23)	33 (31–36)
	Living in someone else’s home	41 (33–49)	47 (43–51)	52 (44–59)	64 (59–69)	37 (24–50)	53 (45–62)	30 (25–36)	46 (43–50)	38 (21–54)	39 (28–49)	24 (15–33)	35 (28–42)
Perception of adequate housing	Suitable for confinement	24 (22–26)	34 (33–35)	41 (39–43)	56 (54–58)	33 (30–37)	43 (41–46)	26 (24–28)	38 (37–40)	25 (19–30)	34 (29–38)	16 (14–18)	29 (28–31)
	Not suitable for confinement	41 (37–46)	50 (48–53)	55 (49–61)	72 (67–76)	48 (38–59)	63 (56–70)	38 (34–43)	52 (49–55)	57 (43–70)	53 (45–61)	33 (26–39)	50 (45–54)
Household composition	Living alone	25 (21–28)	29 (27–31)	43 (36–49)	50 (45–55)	33 (23–42)	45 (36–53)	28 (22–33)	40 (35–44)	39 (21–57)	23 (10–35)	19 (15–23)	28 (25–31)
	Living with other people	28 (26–30)	38 (37–39)	43 (41–46)	59 (57–61)	36 (32–39)	46 (43–49)	28 (26–30)	41 (40–43)	30 (24–35)	39 (35–43)	18 (16–19)	32 (31–33)
Presence of minors in the household,	No	26 (24–29)	36 (34–37)	42 (39–46)	56 (54–59)	38 (32–44)	44 (40–48)	28 (25–31)	40 (37–42)	29 (20–38)	32 (25–38)	21 (16–25)	33 (30–37)
	Yes	29 (26–32)	41 (40–43)	44 (41–48)	61 (59–64)	34 (29–38)	47 (44–51)	29 (26–31)	43 (41–45)	30 (23–37)	43 (38–48)	22 (17–26)	35 (32–38)
Presence of dependents in the household	No	25 (22–29)	34 (32–35)	40 (37–43)	54 (51–56)	33 (28–38)	42 (38–46)	26 (23–28)	37 (35–39)	24 (16–32)	35 (29–41)	24 (17–32)	35 (30–39)
	Yes	28 (26–31)	40 (39–42)	48 (44–52)	65 (62–67)	38 (33–44)	49 (45–52)	32 (29–35)	46 (43–48)	34 (26–42)	42 (37–47)	26 (18–34)	42 (37–48)
Household work	Other persons/equally among household members	26 (24–29)	35 (34–37)	42 (39–44)	55 (53–58)	35 (31–39)	43 (40–47)	27 (25–29)	38 (37–40)	29 (23–35)	37 (32–41)	NA	NA
	Mostly by myself	33 (28–38)	41 (40–43)	55 (48–61)	64 (61–66)	40 (28–52)	51 (47–56)	38 (32–45)	47 (45–50)	35 (16–53)	45 (38–52)	NA	NA
Concern about living with household members	Nothing or little	15 (12–18)	27 (25–29)	33 (30–37)	49 (47–52)	27 (22–32)	34 (30–38)	19 (16–21)	32 (30–34)	23 (15–31)	27 (21–33)	12 (11–14)	26 (24–27)
	Moderate, quite a bit or a lot	33 (31–36)	44 (42–45)	52 (49–55)	69 (67–72)	42 (37–47)	56 (52–59)	37 (34–40)	51 (49–53)	34 (27–41)	46 (41–51)	38 (32–43)	53 (50–56)
Concern about schooling	Nothing or little	27 (19–34)	37 (34–41)	41 (35–47)	59 (55–63)	24 (12–36)	37 (27–47)	29 (23–34)	37 (33–41)	25 (11–39)	28 (19–37)	16 (12–21)	29 (27–32)
	Moderate, quite a bit or a lot	30 (26–33)	43 (41–45)	46 (41–50)	64 (61–67)	35 (30–40)	49 (45–53)	32 (29–35)	45 (43–47)	30 (22–38)	48 (42–54)	25 (20–30)	40 (37–43)

NA: not applicable (no information on this variable). ^1^ Race/skin color self-declaration in case of Brazil (no: white; yes: black, brown, yellow, or indigenous).

**Table 4 ijerph-20-05722-t004:** Prevalence of depressive symptoms according to gender and sociodemographic characteristics in Brazil, Chile, Ecuador, Mexico, Peru, and Spain during the first wave of COVID-19 lockdown.

		Brasil		Chile		Ecuador		Mexico		Perú		Spain	
		Men	Women	Men	Women	Men	Women	Men	Women	Men	Women	Men	Women
		N = 2414	N = 9323	N = 1799	N = 3164	N = 709	N = 1493	N = 2208	N = 4865	N = 298	N = 647	N = 2074	N = 5107
		Prevalence (IC95%)	Prevalence (IC95%)	Prevalence (IC95%)	Prevalence (IC95%)	Prevalence (IC95%)	Prevalence (IC95%)	Prevalence (IC95%)	Prevalence (IC95%)	Prevalence (IC95%)	Prevalence (IC95%)	Prevalence (IC95%)	Prevalence (IC95%)
Depressive symptomns		28 (26–30)	37 (36–38)	31 (29–33)	42 (40–44)	32 (28–35)	43 (40–45)	28 (26–30)	40 (39–42)	27 (21–32)	37 (33–40)	16 (15–18)	28 (27–30)
Educational level	Basic or bachelor’s degree	36 (32–40)	47 (45–50)	46 (40–52)	60 (55–64)	44 (32–55)	61 (54–68)	31 (22–41)	40 (32–47)	37 (21–53)	58 (48–67)	18 (15–22)	35 (32–39)
	Technical studies	34 (27–41)	49 (45–54)	30 (25–35)	43 (39–48)	31 (13–49)	42 (29–56)	36 (32–40)	48 (45–51)	46 (31–62)	40 (31–50)	18 (13–23)	32 (28–36)
	University studies	25 (23–27)	34 (33–35)	28 (26–31)	38 (36–40)	30 (27–34)	40 (38–43)	26 (23–28)	38 (36–39)	22 (16–27)	31 (27–35)	16 (14–18)	26 (25–28)
Age	18–34	45 (42–49)	54 (52–56)	46 (41–50)	56 (53–59)	40 (35–46)	49 (46–53)	38 (35–41)	51 (49–53)	40 (31–49)	46 (40–51)	28 (24–32)	42 (40–45)
	35–64	23 (21–25)	33 (32–34)	28 (25–30)	36 (34–39)	26 (22–31)	36 (32–39)	21 (19–24)	31 (29–33)	18 (13–24)	29 (24–34)	14 (12–16)	24 (23–26)
	≥65	8 (5–11)	18 (16–21)	7 (3–12)	25 (18–31)	25 (8–42)	22 (3–41)	7 (2–12)	16 (10–23)	23 (0–46)	05 (0–13)	06 (3–9)	13 (9–17)
Indigenous group membership ^1^	No	27 (25–29)	36 (34–37)	30 (23–36)	41 (40–43)	32 (29–36)	42 (40–45)	28 (26–30)	40 (39–42)	25 (20–30)	36 (32–40)	NA	NA
	Yes	30 (27–33)	40 (39–42)	32 (29–34)	47 (42–53)	27 (16–39)	50 (40–59)	25 (19–31)	41 (37–45)	36 (22–50)	43 (32–54)	NA	NA
Country of origin	Born in the country of study	28 (27–30)	37 (36–38)	32 (30–34)	38 (36–39)	33 (29–36)	43 (40–46)	28 (26–30)	40 (39–42)	NA	NA	16 (15–18)	28 (27–29)
	Migrant	16 (06–25)	29 (22–36)	19 (11–27)	34 (27–42)	15 (04–27)	35 (22–47)	21 (9–32)	40 (29–50)	NA	NA	19 (14–24)	31 (27–35)
Pre-pandemic employment status,	Employed	28 (25–30)	37 (36–39)	30 (27–32)	39 (37–41)	30 (26–33)	38 (35–41)	26 (24–29)	38 (36–40)	23 (18–28)	31 (27–35)	16 (14–18)	26 (25–27)
	Unemployed	31 (27–34)	38 (36–39)	39 (33–46)	52 (48–56)	43 (34–51)	56 (51–61)	32 (28–36)	45 (43–47)	46 (32–60)	52 (45–59)	18 (15–21)	34 (32–36)
Change in employment status during the pandemic	No change/improvement in employment condition	19 (17–21)	29 (28–30)	25 (22–28)	35 (32–37)	22 (16–27)	34 (29–38)	22 (19–24)	32 (30–34)	19 (12–25)	28 (23–34)	13 (11–14)	24 (22–25)
	Worsening in employment condition	37 (34–39)	45 (44–47)	36 (33–39)	48 (46–50)	36 (32–41)	46 (43–49)	34 (32–37)	48 (46–50)	33 (25–41)	41 (35–47)	22 (19–25)	35 (33–37)
Housing tenure	Own home	25 (23–27)	34 (32–35)	27 (24–30)	37 (34–39)	31 (27–35)	40 (37–43)	24 (22–26)	36 (34–38)	22 (16–28)	34 (29–38)	14 (12–16)	26 (25–27)
	Lease or rent	35 (31–38)	44 (41–46)	35 (32–39)	48 (45–51)	32 (25–38)	48 (43–53)	35 (31–40)	46 (43–49)	35 (25–45)	44 (37–52)	20 (16–23)	32 (29–35)
	Living in someone else’s home	38 (30–46)	51 (47–55)	43 (35–50)	52 (46–57)	38 (25–51)	49 (40–57)	35 (29–41)	50 (46–53)	26 (10–41)	35 (24–45)	26 (16–35)	40 (32–47)
Perception of adequate housing	Suitable for confinement	25 (23–27)	34 (33–35)	29 (27–31)	39 (38–41)	30 (27–34)	41 (38–44)	26 (24–28)	37 (36–39)	23 (18–28)	33 (28–37)	14 (13–16)	26 (25–27)
	Not suitable for confinement	40 (36–45)	52 (50–55)	45 (39–51)	62 (57–67)	43 (33–54)	54 (47–61)	37 (33–42)	52 (49–55)	43 (29–56)	49 (41–57)	34 (28–41)	48 (44–52)
Household composition	Living alone	27 (23–31)	31 (29–33)	33 (27–40)	38 (33–43)	31 (22–41)	49 (41–58)	29 (24–35)	42 (37–46)	39 (21–57)	23 (11–36)	19 (15–23)	30 (27–33)
	Living with other people	28 (26–30)	38 (37–39)	31 (29–33)	43 (41–44)	32 (28–36)	42 (39–45)	28 (26–30)	40 (39–42)	25 (20–30)	38 (34–41)	16 (14–18)	28 (27–29)
Presence of minors in the household,	No	28 (25–30)	37 (35–38)	33 (30–37)	44 (42–47)	36 (30–41)	46 (42–50)	29 (27–32)	38 (36–40)	26 (18–35)	33 (26–39)	15 (11–19)	27 (24–30)
	Yes	29 (26–32)	40 (39–42)	29 (26–32)	41 (38–43)	29 (24–34)	39 (36–43)	27 (24–29)	42 (40–44)	25 (18–31)	40 (35–45)	15 (11–19)	26 (23–28)
Presence of dependents in the household	No	29 (25–33)	36 (34–37)	28 (25–30)	39 (37–42)	30 (25–35)	40 (36–44)	23 (20–25)	37 (35–39)	22 (14–29)	37 (31–43)	25 (18–32)	29 (25–33)
	Yes	28 (26–31)	40 (38–41)	36 (32–40)	47 (44–50)	34 (28–39)	44 (40–47)	25 (22–28)	43 (41–45)	28 (21–35)	38 (33–43)	17 (10–25)	28 (23–33)
Household work	Other persons/equally among household members	26 (24–29)	37 (35–38)	29 (27–31)	40 (38–43)	32 (28–35)	42 (39–45)	27 (25–29)	39 (38–41)	24 (19–30)	36 (31–40)	NA	NA
	Mostly by myself	39 (34–44)	40 (38–41)	45 (38–52)	45 (43–48)	35 (23–46)	43 (38–47)	38 (32–45)	42 (40–45)	35 (16–53)	42 (35–49)	NA	NA
Concern about living with household members	Nothing or little	19 (16–23)	30 (28–31)	24 (21–27)	33 (31–36)	27 (22–32)	32 (28–36)	21 (18–23)	34 (32–36)	19 (11–27)	27 (21–33)	12 (10–13)	22 (20–23)
	Moderate, quite a bit or a lot	32 (30–35)	42 (41–44)	37 (34–40)	53 (50–55)	36 (31–41)	50 (47–54)	35 (32–38)	47 (45–49)	29 (22–36)	44 (39–49)	32 (27–37)	47 (44–50)
Concern about schooling	Nothing or little	28 (21–36)	35 (32–39)	28 (22–33)	37 (33–41)	20 (09–31)	27 (18–36)	27 (21–32)	37 (33–41)	31 (16–45)	30 (21–39)	12 (8–16)	19 (16–21)
	Moderate, quite a bit or a lot	29 (26–33)	42 (40–43)	29 (25–33)	44 (40–47)	31 (25–36)	41 (37–45)	27 (23–30)	44 (42–46)	23 (15–30)	45 (39–50)	17 (13–21)	33 (30–37)

NA: not applicable (no information on this variable). ^1^ Race/skin color self-declaration in case of Brazil (no: white; yes: black, brown, yellow, or indigenous).

## Data Availability

Data cannot be shared publicly because of ethical restrictions. The Ethical Committee does not allow us to share the data publicly as our data contain sensitive personal information and cannot be fully anonymized. Data are available from the Research Ethics Committee of the Institut de Recerca en Atenció Primària Jordi Gol i Gurina (IDIAPJGol) (contact via cei@idiapjgol.info) for researchers who meet the criteria for access to confidential data.

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
