# Peer review of "Social Inequalities in Mental Health and Self-Perceived Health in the First Wave of COVID-19 Lockdown in Latin America and Spain: Results of an Online Observational Study"

_ijerph, 2023, doi:10.3390/ijerph20095722_

Round 1
Reviewer 1 Report
Dear authors,
Thank you for your study on the impact of COVID-19 lockdowns on mental health and self-perceived health in Latin America and Spain. While the study provides valuable insights into the issue, I have some concerns that I hope you can address in your revisions. Specifically, I have stated 8 areas you need to consider in your revisions.
First, the introduction could benefit from (1) a more concise and focused research question or objective that explicitly states what the study aims to achieve. (2) Providing a clearer and more structured overview of the research design, methods, and data sources would also be helpful in understanding how the study was conducted and what kind of data was used. Moreover, please (3) consider using more specific and concrete examples to illustrate the impact of the pandemic on mental health and self-perceived health in the countries under study, and including more explicit references to relevant literature.
Regarding the presentation of your descriptive statistics, while you have provided a wealth of information on various social and economic factors, I feel that the data is presented in a manner that does not highlight the key findings and how they relate to your research objectives and questions. It would be helpful (4) to provide a more concise summary of your key findings and how they support your study's goals. Plus, (5) organizing your descriptive statistics in a more cohesive and focused manner, rather than simply listing various statistics and values without context, would make it easier for readers to understand the significance of your findings and (6) how they contribute to the larger discussion of mental health and self-perceived health during the pandemic.
In the discussion section, while the authors highlight the importance of social determinants of mental health in Latin America and Spain, please consider discussing the limitations of the study, especially (7) due to the potential for selection bias and the self-reported nature of the data. (8) Providing suggestions for future research, such as longitudinal studies to track the impact of the pandemic on mental health over time, would also be beneficial.
Thank you for considering my feedback.
Reviewer 2 Report
Dear authors, congratulations on such an extensive sample! The comparison of different countries is of outmost relevance for pub heath policy makers.
However, the research questions or hypothesis should be more clearly presented and the discussions/conclusions should reflect them.
The conclusions should be improved in terms of presenting how the results obtained can contribute to better social policies or by indicating specific suggestions for improving mental health policies.
Reviewer 3 Report
The study is impressively extensive and comprehensive, and its findings are rich and valuable. I found the differences among the population groups that do not match the other countries' same categories (e.g., Chilean indigenous men and Peruvian indigenous women) facinating, and these data, among others, provide a rich food for thought.
What I found lacking in the present manuscript are: 1. Theoretical contextualization; and 2. Rigorous contexutalization of the findings. The manuscript could use a robust literature review section after "1. Introduction" and "2. Materials and Methods." As a reader, I would have liked to learn about comparable studies in other regions/countries about a similar theme, and what they have found. While the "4. Discussion" section frequently makes references to the existing studies and their findings, there is not much theoretical and intellectual background to situate this study's findings. Relatedly, the brief discussion ("4.1. Limitations and strengths") could use a bit more elaboration on how this particular finding, beyond the regional "gap" in the existing studies, can add to the exiting body of theoretical and intellectual conversation on the pandemic's impact on SPH and mental health, which requires a more robust theoretical "set-up" in the earlier part of the manuscript. As for the discussion of findings, I found it wanting, because much of the contexutalizing commentaries are vague (e.g., "There were potlitical instabilities in X and Y countries") and no discussion on how, exactly, these political and socio-economic determinants of health might have impacted the data. While the authors may not be able to provide detailed anaysis for every single significant data point, it would be helpful to offer a more nuanced and precise linkage between the found data and potential unique contributing factors.
Round 2
Reviewer 1 Report
The authors seem to have put in significant effort to revise the paper in response to the previous feedback.
Reviewer 3 Report
The additional sections that provide socio-political contexts of the countries studied helped readers understand the authors' discussions of the findings better. While the analysis of the findings in the disucssion section still feels a bit too speculative, the authors made a concerted effort to link the data with the given pandemic context of each country.